# Control What You Can
# Intrinsically Motivated Task-Planning Agent

**Sebastian Blaes**       **Marin Vlastelica  Pogančić**       **Jia-Jie Zhu**       **Georg Martius**

Autonomous Learning Group
Max Planck Institute for Intelligent Systems
Tübingen, Germany
{sebastian.blaes,marin.vlastelica,jzhu,georg.martius}@tue.mpg.de

## Abstract

We present a novel intrinsically motivated agent that learns how to control the environment in a sample efficient manner, that is with as few environment interactions as possible, by optimizing learning progress. It learns what can be controlled, how to allocate time and attention as well as the relations between objects using surprise-based motivation. The effectiveness of our method is demonstrated in a synthetic and robotic manipulation environment yielding considerably improved performance and smaller sample complexity compared to an intrinsically motivated, non-hierarchical and state-of-the-art hierarchical baseline. In a nutshell, our work combines several task-level planning agent structures (backtracking search on task-graph, probabilistic road-maps, allocation of search efforts) with intrinsic motivation to achieve learning from scratch.

## 1   Introduction

This paper studies the question of *how to make an autonomous agent learn to gain maximal control of its environment* under little external reward. To answer this question, we turn to the true learning experts: children. Children are remarkably fast in learning new skills; How do they do this? In recent years psychologists and psychiatrists acquired a much better understanding of the underlying mechanisms that facilitate these abilities. Babies seemingly do it by conducting experiments and analyzing the statistics in their observations to form intuitive theories about the world [12]. Thereby, human learning seems to be supported by hard-wired abilities, e.g. identifying human faces or the well tuned relationship between teacher (adult) and student (infant) [11] and by self-motivated playing, often with any objects within their reach. The purpose may not be immediately clear to us. But to play is to manipulate, to *gain control*. In the spirit of this cognitive developmental process, we specifically design an agent that is 1) intrinsically motivated to gain control of the environment 2) capable of learning its own curriculum and to reason about object relations.

As a motivational example, consider an environment with a heavy object that cannot be moved without using a tool such as a forklift, as depicted in Fig. 1(**6**). To move the heavy object, the agent needs to learn first how to control itself and then how to use the tool. In the beginning, we do not assume the agent has knowledge of the tool, object, or physics. Everything needs to be learned *from scratch* which is highly challenging for current RL algorithms. Without external rewards, an agent may be driven by intrinsic motivation (IM) to gain control of its own internal representation of the world, which includes itself and objects in the environment. It often faces a decision of what to attempt to learn with limited time and attention: If there are several objects that can be manipulated, which one should be dealt with first? In our approach the scheduling is solved by an automatic curriculum that aims at improving *learning progress*. The learning progress may have its unique advantage over other

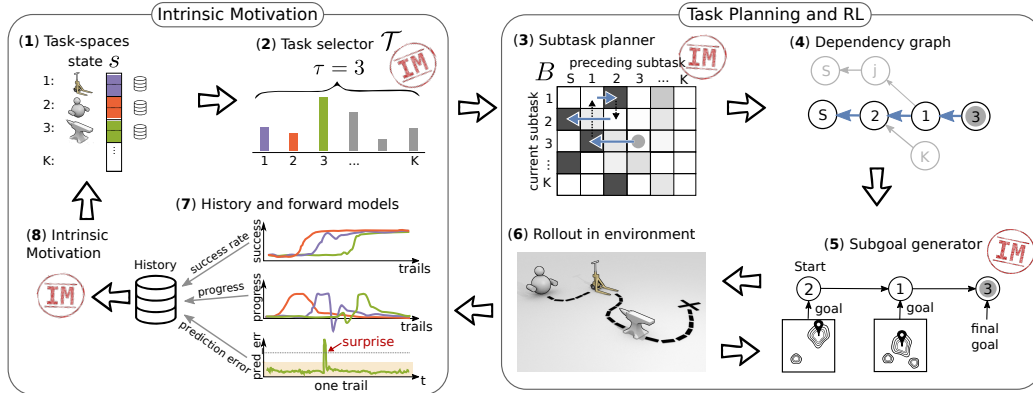

Figure 1: Overview of CWYC method. All components except (1) are learned. Details can be found in the main text. Videos and code are available at https://s-bl.github.io/cwyc/

quantities such as prediction error (curiosity): It renders unsolvable tasks uninteresting as soon as progress stalls.

Instead of an end-to-end architecture, we adopt a core *reasoning structure* about tasks and sub-goals. Inspired by task-level planning methods from the robotics and AI planning communities, we model the agent using a planning architecture in the form of chained sub-tasks. In practice, this is modeled as a task-graph as in Fig. 1(**4**). In order to manipulate the state of the tool, the agent and the tool need to be in a specific relation. Our agent learns such relationships by an attention mechanism bootstrapped by surprise detection.

Our main contributions are:

1. We propose to use *maximizing controllability* and *surprise* as intrinsic motivation for solving challenging control problems. The computational effectiveness of this cognitive development-inspired approach is empirically demonstrated.

2. We propose to adopt several *task-level planning ideas* (backtracking search on task-graph/goal regression, probabilistic road-maps, allocation of search efforts) for designing IM agents to achieve task completion and skill acquisition from scratch.

To our knowledge, no prior IM study has adopted similar controllability and task-planning insights. The contributions are validated through 1) a synthetic environment, with exhaustive analysis and ablation, that cannot be solved by state-of-the-art methods even with oracle rewards; 2) a robotic manipulation environment where tool supported manipulation is necessary.

## 2 Related Work

In this section, we give a survey on the recent computational approaches to intrinsic motivation (IM). This is by no means comprehensive due to the large body of literature on this topic. Generally speaking, there are a few types of IM in literature: *learning progress (competence, empowerment), curiosity (surprise, prediction error), self-play (adversarial generation), auxiliary tasks, maximizing information theoretical quantities*, etc. To help readers clearly understand the relation between our work and the literature, we provide the following table.

| | Intrinsic motivation | Computational methods |
|---|---|---|
| **CWYC Ours** | **learning progress + surprise** | **task-level planning, relational attention** |
| h-DQN [21] | reaching subgoals | HRL, DQN |
| IMGEP [10] | learning progress | memory-based |
| CURIOUS [8] | learning progress | DDPG, HER, E-UVFA |
| SAC-X [29] | auxiliary task | HRL, (DDPG-like) PI |
| Relational RL [38] | - | relation net, IMPALA |
| ICM [26] | prediction error | A3C, ICM |
| Goal GAN [9] | adversarial goal | GAN, TRPO |
| Asymmetric self-play [34] | self-play | Alice/Bob, TRPO, REINFORCE |

*Learning progress* describes the rate of change of an agent gaining competence in certain skills. It is a heuristic for measuring interests inspired by observing human children. This is the focus of many recent studies [2, 8, 10, 18, 25, 31, 32]. Our work can be thought of as an instantiation of that as well using maximizing controllability and a task-planning structure. Empowerment [20] proposes a quantity that measures the control of the agent over its future sensory input. *Curiosity*, as a form of IM, is usually modeled as the prediction error of the agent's world model. For example, in challenging video game domains it can lead to remarkable success [26] or to learn options [7]. *Self-play* as IM, where two agents engage in an adversarial game, was demonstrated to increase learning speed [34]. This is also related to the idea of using GANs for goal generation as in [9].

Recently, *auxiliary prediction tasks* were used to aid representation learning[17]. In comparison, our goal is not to train the feature representation but to study the developmental process. Similarly, *informed auxiliary tasks* as a form of IM was considered in [29]. Many RL tasks can be formulated as aiming to maximize certain *information theoretical quantities* [14, 23, 36, 37]. In contrast, we focus on IM inspired by human children. In [21] a list of given sub-goals is scheduled whereas our attention model/goal generation is learned. Our work is closely related to [38], multi-head self-attention is used to learn non-local relations between entities which are then fed as input into an actor-critic network. In this work, learning these relations is separated from learning the policy and done by a low capacity network. More details are given in Sec. 3. In addition, we learn an automatic curriculum.

Task-level planning has been extensively studied in robotics and AI planning communities in form of geometric planning methods (e.g., RRT [22], PRM [19]) and optimization-based planning [28, 33]. There is a parallel between the sparse reward problem and optimization-based planning: the lack of gradient information if the robot is not in contact with the object of interest. Notably, our use of the surprise signals is reminiscent to the event-triggered control design [3, 16] in the control community and was also proposed in cognitive sciences [6].

## 3    Method

We call our method *Control What You Can* (CWYC). The goal of this work is to make an agent learn to control itself and objects in its environment, or more generically, to control the components of its internal representation. We assume the observable state-space $\mathcal{S} \in \mathbb{R}^n$ is partitioned into groups of potentially controllable components (coordinates) referred to as *goal spaces*, similar to [1]. Manipulating these components is formulated as self-imposed tasks. These can be the agent's position (task: move agent to location $(x, y)$), object positions (task: move object to position $(x, y)$), etc. The component's semantics and whether it is controllable are unknown to the agent. The perception problem, that is constructing goal spaces from high-dimensional image data or other sensor modalities, is an orthogonal line of research. Readers interested in representation learning are referred to, e. g. [5, 27]. Formally, the coordinates in the state $s$, corresponding to each goal-reaching task $\tau \in \mathcal{K} = \{1, 2, \ldots, K\}$, are indexed by $m_\tau \subset \{1, 2, \ldots, n\}$ and denoted by $s_{m_\tau}$. The *goal* in each task is denoted as $g_\tau \in \mathbb{R}^{|m_\tau|}$. For instance, if task 1 has its goal-space along the coordinates 3 and 4 (e.g. agent's location) then $m_1 = (3, 4)$ and $s_{m_1} = (s_3, s_4)$ are corresponding state values. It is assumed that goal spaces are non-overlapping and encompass only simultaneously controllable components.

During the learning/development phase, the agent can decide which task (e. g. itself or object) it attempts to control. Intuitively, it should be beneficial for the learning algorithm to concentrate on tasks where the agent can make progress in and inferring potential task dependencies. In order to express the capability of controlling a certain object, we consider goal-reaching tasks with randomly selected goals. When the agent can reach any goals allowed by the environment, it has achieved control of the component (e. g. moving a box to any desired location). In many challenging scenarios, e. g. object manipulation, there are "funnel states" that must be discovered. For instance, in a tool-use task the funnel states are where the agent picks up the tool and where the tool touches another object that needs to be manipulated. Our architecture combines relational learning embedded in an overall intrinsically motivated learning framework based on a learned probabilistic graph that chains low-level goal-directed RL controllers.

Our approach contains several components as illustrated in Fig. 1. Their detailed interplay is as follows: The tasks (**1**) control groups of components (coordinates) of the state. A task selector (bandit)(**2**) is used to select a self-imposed task $\tau$ (final task) maximizing expected learning progress.

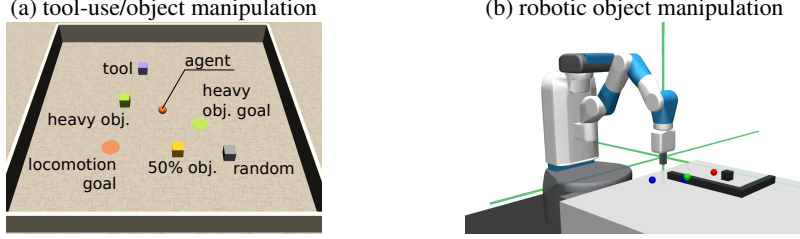

(a) tool-use/object manipulation      (b) robotic object manipulation

Figure 2: Environments used to test CWYC. (a) basic tool-use/object manipulation environment; (b) robotic object manipulation environment. The hook needs to be used to move the box.

Given a final task, the task planner (**3**) computes a viable sub-task sequence (bold) from a learned task graph (**4**). The sub-goal generators (**5**) (relational attention networks) create for every time step a goal $g_i(t)$ in the current sub-task $i$. The goal-conditioned low-level policies for each task control the agent $a(t) \sim \pi_i(s(t), g_i(t))$ in the environment (**6**). Let us comprise (**3,4,5**, $\pi_i \forall i$) into the acting policy $a(t) \sim \Pi(s(t), \tau, g_\tau)$ (internally using the current sub-task policy and goal etc). After one rollout different quantities, measuring the training progress, are computed and stored in the per task history buffer (**7**). An intrinsic motivation module (**8**) computes the rewards and target signals for (**2**), (**3**), and (**5**) based on learning progress and prediction errors. All components are trained concurrently and without external supervision. Prior knowledge enters only in the form of specifying the goal spaces (groups of coordinates of the state space). The environment allows the agent to select which task to do next and generates a random arrangement with a random goal.

## 3.1 Intrinsic Motivation

In general, our agent is motivated to learn as fast as possible, i. e. maximizing instantaneous learning progress, and to be as successful as possible, i.e. maximizing success rate, in each task. When performing a particular task $\tau$, with the goal $g_\tau$ the agent computes the reward for the low-level controller as the negative distance to the goal as $r_i(t) = -\|s_{m_i}(t) - g_i\|^2$ and declares success as: $\mathrm{succ}_i = \max_t \llbracket \|s_{m_i}(t) - g_i\|^2 \leq \delta_i \rrbracket$ where $\delta_i$ is a precision threshold and $\llbracket \cdot \rrbracket$ is the Iverson bracket. The maximum is taken over all time steps in the current rollout (trial). We choose the euclidean distance between the task relevant sub-state to the goal state as distance metric because it is general in that it does not impose a particular structure on the goal spaces and therefore can be easily applied to any goal space. We calculate the following key measures to quantify intrinsic motivations:

**Success rate (controlability)** $\mathrm{sr}_i = \mathbb{E}_\eta(\mathrm{succ}_i)$, where $\eta$ is the state distribution induced by $\Pi(\cdot, i, \cdot)$. In practice $\mathrm{sr}_i$ is estimated as a running mean of the last attempts of task $i$.

**Learning progress** $\rho_i = \frac{\Delta \, \mathrm{sr}_i}{\Delta \, \mathrm{rollout}}$ is the time derivative of the success rate, quantifying whether the agent gets better at task $i$ compared to earlier attempts.

Initially, any success signals might be so sparse that learning becomes slow because of uninformed exploration. Hence, we employ *surprise* as a proxy that guides the agent's attention to tasks and states that might be *interesting*.

**Prediction error** $e_i(t)$ in goal space $i$ of a forward model $f : [\mathcal{S} \times \mathcal{A}] \rightarrow \mathcal{S}$ trained using squared loss $e(t) = \|(f(s(t), a(t)) + s(t)) - s(t+1)\|^2$ and $e_i = e_{m_i}$ denotes the error in the goal space $i$.

**Surprising events** $\mathrm{surprise}_i(t) \in \{0, 1\}$ is 1 if the prediction error $e_{m_i}(t)$ in task $i$ exceeds a confidence interval (computed over the history), 0 otherwise, see also [13].

To understand why surprising events can be informative, let us consider again our example: Assume the agent just knows how to move itself. It will move around and will not be able to manipulate other parts of its state-space, i. e. it can neither move the heavy box nor the tool. Whenever it accidentally hits the tool, the tool moves and creates a surprise signal in the coordinates of the tool task. Thus, it is likely that this particular situation is a good starting point for solving the tool task and make further explorations.

## 3.2 Task-Planning Architecture

The task selector $\mathcal{T}$, Fig. 1(**2**) models the learning progress when attempting to solve a task. It is implemented as a multi-armed bandit. While no learning progress is available, the surprise signal is used as a proxy. Thus, the internal reward signal for the bandit for a rollout attempting task $i$ is

$$r_i^{\mathcal{T}} = |\rho_i| + \beta^{\mathcal{T}} \max_t (\text{surprise}_i(t)) \tag{1}$$

with $\beta^{\mathcal{T}} \ll 1$. The multi-armed bandit is used to choose the (final) task for a rollout using a stochastic policy. More details can be found in Sec. A.1. In our setup, the corresponding goal within this task is determined by the environment (in a random fashion).

Because difficult tasks require sub-tasks to be performed in certain orders, a task planner determines the sequence of sub-tasks. The task planner models how quick (sub)-task $i$ can be solved when performing sub-task $j$ directly before it. As before, we use surprising events as a proxy signal for potential future success. The values of each task transition is captured by $B_{i,j}$, where $i \in [1, \dots, K]$ and $j \in [S, 1, \dots, K]$ with $S$ representing the "start":

$$B_{i,j} = \frac{Q_{i,j}^B}{\sum_k Q_{i,k}^B} \quad \text{with} \quad Q_{i,j}^B = \left\langle 1 - \frac{T_{i,j}}{T^{\max}} + \beta^B \max_t (\text{surprise}_i(t)) \right\rangle \tag{2}$$

where $\langle \cdot \rangle$ denotes a running average and $T_{i,j}$ is the runtime for solving task $i$ by doing task $j$ before (maximum number time steps $T^{\max}$ if not successful). Similarly to Eq. 1, this quantity is initially dominated by the surprise signals and later by the actual success values.

The matrix $B$ represents the adjacency matrix of the task graph, see Fig. 1(**4**). It is used to construct a sequence of sub-tasks by starting from the final task $\tau$ and determining the previous sub-task with an $\epsilon$-greedy policy using $B_{\tau,\cdot}$. Then this is repeated for the next (prerequisite) sub-task, until $S$ (start) is sampled (no loops are allowed), see also Fig. 1(**3**) and (**4**).

Each (sub)-task is itself a goal-reaching problem. In order to decide which sub-goals need to be chosen we employ an attention network for each task transition, i.e. $G_{i,j}$ for the transition from task $j$ to task $i$. As before, the aim of the goal proposal network $G_{i,j}$ is to maximize the success rate of solving task $i$ when using the proposed goal in task $j$ before. In the example, in order to pick up the tool, the goal of the preceding locomotion task should be the location of the tool. An attention network that can learn relations between observations is required. We use an architecture that models local pairwise distance relationships. It associates a value/attention to each point in the goal-space of the preceding task as a function of the state $s$: $G_{i,j} : \mathcal{S} \to \mathbb{R}$: (omitting index $_{i,j}$)

$$G_{i,j}(s) = e^{-\gamma \sum_{k=1}^n \sum_{l=k+1}^n \|w_{kl}^1 s_k + w_{kl}^2 s_l + w_{kl}^3\|^2} \tag{3}$$

where $w^1$, $w^2$, $w^3$, and $\gamma$ are trainable parameters. The network is trained by regression, minimizing the loss

$$\mathcal{L}_{i,j}^G(w_{kl}^1, w_{kl}^2, w_{kl}^3, \gamma) = \min_{w_{kl}^1, w_{kl}^2, w_{kl}^3, \gamma} \sum_{k=1}^n \|G_{i,j}(s_k) - r_{i,j}^G(s_k)\|^2 \tag{4}$$

where $k$ enumerates training samples and with the following target signal $r_{i,j}^G(s_t) \in [0, 1]$:

$$r_{i,j}^G(s_t) = \min(1, \text{succ}_i \cdot \Gamma_{i,j}(s_t) + \text{surprise}_i(t)) \tag{5}$$

for all $s_t$ that occurred during task $j$ where $\Gamma_{i,j}(s)$ is 1 if the agent decides to switch from task $j$ to task $i$ in state $s$ and zero otherwise. To get an intuition about the parametrization, consider a particular pair of coordinates $(k, l)$, say agent's and tool's $x$-coordinate. The model can express with $w_{k,l}^1 = -w_{k,l}^2 \neq 0$ that both have to be at distance zero for $r^G$ to be 1. However, with $w^3$ the system can also model offsets, global reference points and other relationships. Further details on the architecture and training can be found in Suppl. A.5. We observe that the goal proposal network can learn a relationship after a few examples (in the order of 10), possibly due to the restricted model class. The goal proposal network can be thought of as a relational network [30], albeit is easier to train. Sampling a goal from the network is done by computing the maximum analytically as detailed in Suppl. A.3. The low-level control in each task has its own policy $\pi_i$ learned by soft actor critic (SAC) [15] or DDPG+HER [1].

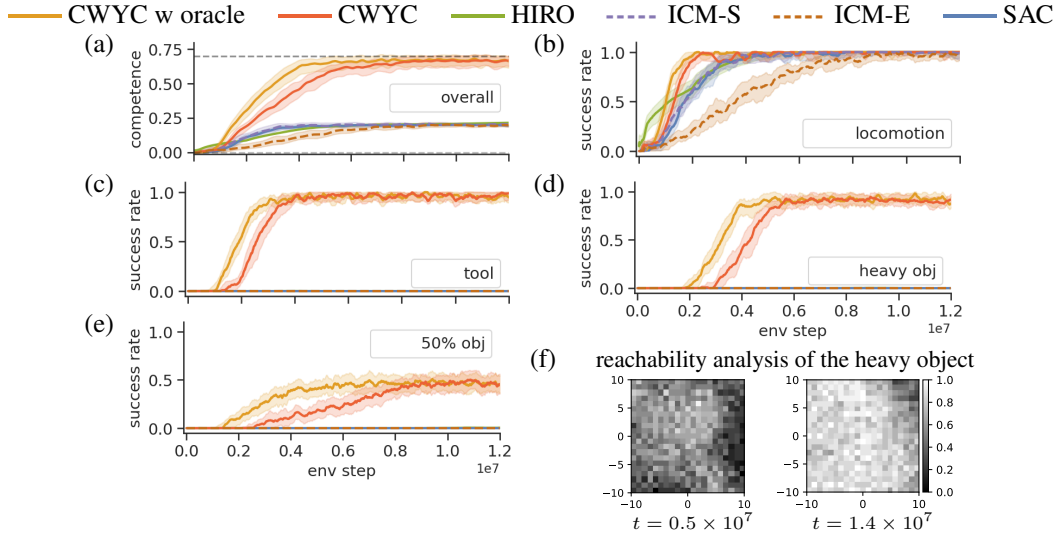

Figure 3: Competence of the agents in controlling all aspects of the synthetic environment. Overall performance (a) (maximal 70%). Individual task competence in (b-e). *HIRO* and *SAC* can only learn the locomotion task. All performance plots (as well in remaining figures) show median and shaded 25% / 75% pecentiles averaged over 10 random seeds. In (c-e) the green curve is below the blue curve. (f) shows the gain in reachability: probability of reaching the point with the heavy box from 20 random starting states. The reachability is initially zero.

## 4 Experimental Results

Through experiments in two different environments, we wish to investigate empirically: does the CWYC agent learn efficiently to gain control over the environment? What about challenging tasks that require a sequence of sub-tasks and uncontrollable objects? How is the behavior of CWYC different from that of other (H)RL agents? To give readers a sense of the computational property of CWYC, we use an implementation[1] of *HIRO* [24] as a HRL baseline which is suitable for continuous control tasks. However, it solves each task independently as it does not support the multi-task setting. As IM baselines, we implement two versions of ICM [26] such that they work with continuous action spaces, using SAC as RL agent. ICM-S uses the surprise signal as additional reward signal while ICM-E uses the raw prediction error directly. In addition, we show the baselines of using only the low-level controllers (*SAC* [15] or *DDPG+HER* [1]) for each individual task independently and spend resources on all tasks with equal probability.

We also add CWYC with a hand-crafted oracle task planner ($B$) and oracle sub-goal generator ($G$) denoted as *CWYC w oracle*, see Suppl. D.1. The code as well as the environment implementations is publicly available. The pseudocode is provided in Suppl. B.

**Synthetic environment.** The synthetic object manipulation arena, as shown in Fig. 2(a), consists of a point mass agent with two degrees of freedom and several objects surrounded by a wall. It is implemented in the MuJoCo physics simulator [35] and has continuous state and action spaces. To make the tasks difficult, we consider the case with 4 different objects: 1. the *tool*, that can be picked up easily; 2. the *heavy object* that needs the tool to be moved; 3. an unreliable object denoted as *50% object* that does not respond to control during 50% of the rollouts; and 4. a *random object* that moves around randomly and cannot be manipulated by the agent, see Fig. 2(a). The detail of the physics in this environment can be found in Suppl. C.1.

Figure 3 shows the performance of the CWYC-agent compared to the hierarchical baseline (HIRO), IM baselines (ICM-(S/E), non-hierarchical baseline (SAC) and the hand-crafted upper baseline (oracle). The main measure is competence, i. e. the overall success-rate ($\frac{1}{K}\sum_{i=1}^{K} \mathrm{sr}_i$) of controlling the internal state, i. e. reaching a random goal in each task-space. In this setting an average maximum of 70% success rate can be achieved due to the completely unsolvable "random object" and the "50%

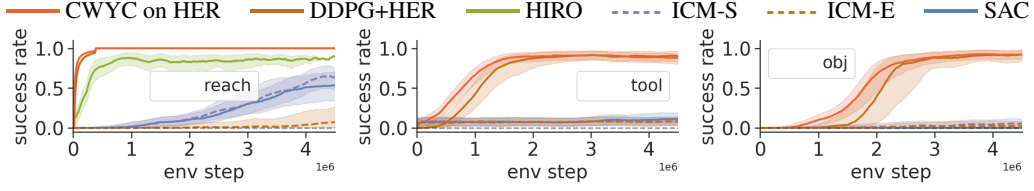

Figure 4: Success rates of reaching, tool using and object manipulation in the robotic environment. CWYC as well as DDPG+HER learn all three tasks perfectly. CWYC has improved sample complexity. SAC, HIRO and ICM-(S/E) learn the reaching task only slowly and the other two tasks not at all.

object" that can be moved only in half of the rollouts and is unmovable in all other rollouts. The results show that our method is able to quickly gain control over the environment, also illustrated by the reachability which grows with time and reaches almost full coverage for the heavy object. After $10^7$ steps[2], the agent can control what is controllable. The *SAC*, ICM-(S/E) and *HIRO* baselines attempt to solve each task independently and spend resources equally between tasks. All four baselines succeed in the locomotion task only. They do not learn to pick up any of the other objects and transport them to a desired location. As a remark, the arena is relatively large such that random encounters are not likely. Providing oracle reward signals makes the baselines (HIRO/SAC) learn to control the tool eventually, but still significantly slower than CWYC, see Fig. 8(b), and the heavy object remains uncontrollable see Suppl. D.2.

**Robotic manipulation.** The robotic manipulation environment consists of a robotic arm with a gripper (3 + 1 DOF) in front of a table with a hook and a box (at random locations), see Fig. 2(b). The box cannot be reached by the gripper directly. Instead, the robot has to use the hook to manipulate the box. The observed state space is 40 dimensional. The environment is based on the OpenAI Gym [4] robotics environment. The goal-spaces/tasks are defined as (1) reaching a target position with the gripper, (2) manipulating the hook, and (3) manipulation the box. Further details can be found in Suppl. C.2. Compared to the synthetic environment, object relations are much less obvious in this environment. Especially the ones involving the hook because of its non-trivial shape. This makes learning object relation much harder. For instance, while trying to grasp the hook, the gripper might touch the hook at wrong positions thus failing at manipulation. However, the objects are relatively close to each other leading to more frequent random manipulations. The results are shown in Fig. 4. Asymptotically, both CWYC and the HER baseline manage to solve all three tasks almost perfectly. The other baselines cannot solve it. Regarding the time required to learn the tasks, our method shows a clear advantage over the HER baseline, solving the 2nd and 3rd task faster.

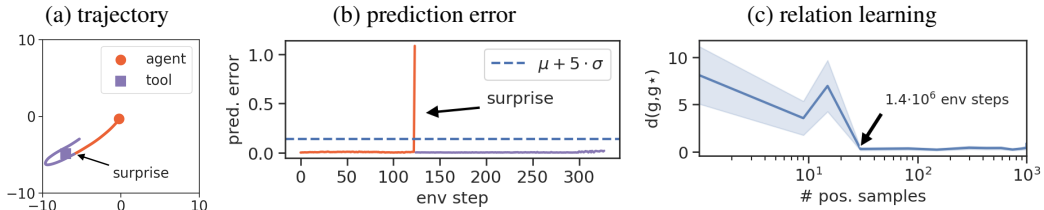

Figure 5: Surprise and relational funnel state learning in the synthetic experiment. (a) the agent bumps into the tool (the red line indicates the agent's trajectory before the encounter, the blue line indicates the joint trajectory of the agent and the tool (after encounter)); (b) surprise (prediction error above $5\sigma$ confidence level) marks the funnel state where tool's and agent's position coincide. (c) avg. distance $\langle d(g, g^\star) \rangle$ of the generated goals $g$ (locomotion targets) from the tool location $g^\star$ in dependence of the number of surprising events.

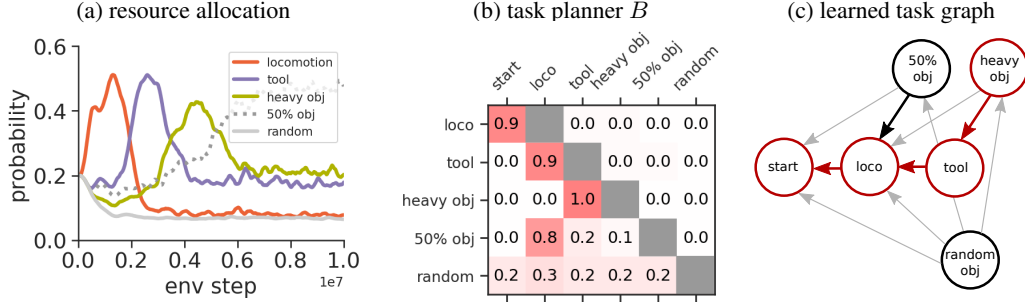

Figure 6: Resource allocation and task planning structure in the synthetic environment. (a) resource allocation (Fig. 1(**2**)): relative time spend on each task in order to maximize learning progress. (b) Task planner (Fig. 1(**3**)): the probabilities $B_{i,j}$ of selecting task $j$ (column) before task $i$ (row). Self-loops (gray) are not permitted. Every sub-task sequence begins in the *start* state. (c) Learned task graph (Fig. 1(**4**)) derived from (b). The arrows point to the preceding task, which corresponds to the planning direction. The red states show an example plan for moving the heavy box.

## 5   Analysis and Ablation: Why and How CWYC Works

How does the agent gain control of the environment? We start by investigating how the **surprising events** help to identify the **funnel states/relationships** – a critical part of our architecture. When the agent is, for instance, involuntarily bumping into a tool, the latter will suddenly move – causing a large prediction error in the forward models of the tool goal-space, see Fig. 5(a,b). Only a few of such surprising observations are needed to make the sub-goal generators, Fig. 1(**5**), effective, see Fig. 5(c). A more detailed analysis follows below. For further details on the training process, see Suppl. A.5.

**Resource allocation** is managed by the task selector, Fig. 1(**2**), based on maximizing learning progress and surprise, see Eq. 1. As shown in Fig. 6(a), starting from a uniform tasks selection, the agent quickly spends most of its time on learning locomotion, because it is the task where the agent makes the most progress in, cf. Fig. 3. After locomotion has been learned well enough, the agent starts to concentrate on new tasks that require the locomotion skill (moving tool and the "50% object"). Afterwards, the heavy object becomes controllable due to the competence in the tool manipulation task (at about $3.5 \cdot 10^6$ steps). The agent automatically shifts its attention to the object manipulation task.

The task selector produces the expected result that simple tasks are solved first and stop getting attention as soon as they cannot be improved more than other tasks. This is in contrast to approaches that are solely based on curiosity/prediction error. When all tasks are controllable (progress plateaus) the 50% object attracts most of the agent's attention due to randomness in the success rate. As a remark, the learned resource allocation of the *oracle* agent is similar to that of CWYC.

Next, we study how the agent **understands the task structure.** The funnel states , discovered above, need to be visited frequently in order to collect data on the indirectly controllable parts of the environment (e. g. tool and heavy box). The natural dependencies between tasks is learned by the

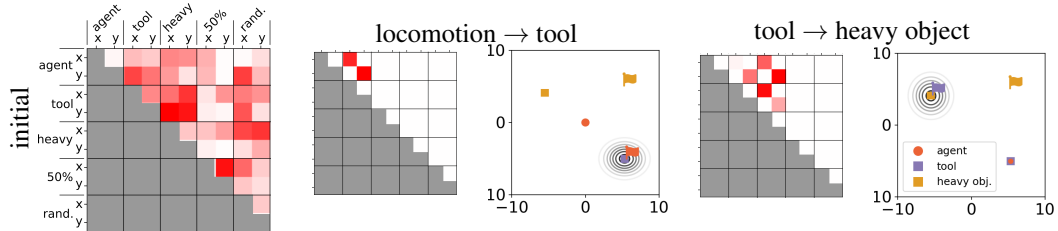

Figure 7: Subgoal proposal networks and the learned object relationships. Intuitively, a nonzero value (red) indicates that this relationship is used. For clarity, each panel shows $\min(|w^1|, |w^2|)$ (see Eq. 3) of the relevant goal proposal networks. Example situations with the corresponding generated goal are presented on the right. The locomotion and tool task goals, i.e red and purple flags, are generated by the subgoal generators, Fig. 1(**5**), the object goal, i.e yellow flag, is sampled randomly.

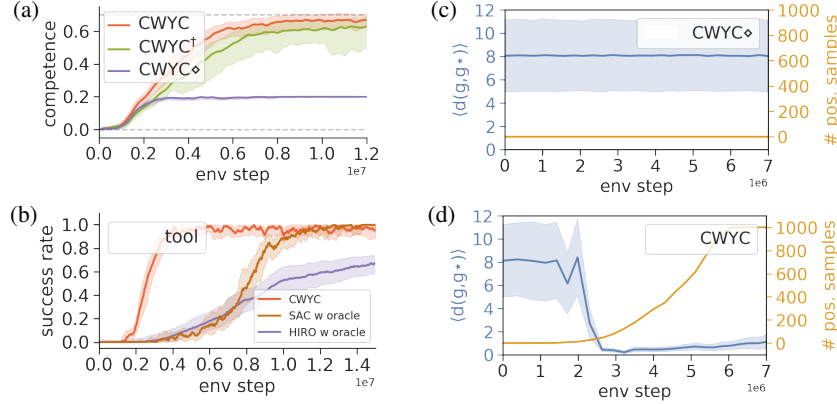

Figure 8: (a) Performance comparison of ablated versions of CWYC: uniform task selection (†) and without surprise signal (◇). (b) baselines with oracle reward on tool task. (c-d) number of positive training samples for goal network and quality of sampled goals (see Fig. 5) CWYC◇ vs. CWYC.

task planner $B$, Fig. 1(**3**). Initially the dependencies between the sub-tasks are unknown such that $B_{i,j} = 0$ resulting in a $1/K+1$ probability of selecting a certain preceding sub-task (or "start"). After learning, the CWYC agent has found which tasks need to be executed in which order , see Fig. 6(b-c). When executing a plan, sub-goals have to be generated. This is where the relational funnel states learned by the sub-goal generators (Fig. 1(**5**)) come in. The sub-goal generators $G_{i,j}$ learn initially from surprising events and attempt to learn the relation among the components of the observation vector. For instance, every time the tool is moved, the agent's location is close to that of the tool.

Figure 7 displays the learned relationships for the sub-goal generation for the locomotion → tool transition and for the tool → heavy object transition. A non-zero value indicates that the corresponding components are involved in the relationship. The full parametrization is visualized and explained in Suppl. A.5. The system identifies that for successfully solving the tool task the coordinates with the agent and the tool have to coincide. Likewise, for moving the heavy box, the agent, tool, and heavy box have to be at the same location. The goal proposal network updates the current goal every 5 steps by computing the goal with the maximal value, see Suppl. A.5 for more details.

We **ablate different components** of our architecture to demonstrate their impact on the performance. We remove the surprise detection, indicated as CWYC◇. A version without resource allocation (uniform task sampling) is denoted as CWYC†. Figure 8(a) shows the performance for the different ablation studies and reveals that the surprise signal is a critical part of the machinery. If removed, it reduces the performance to the SAC baseline, i. e. only solves the locomotion task. Figure 8(c,d) provide insight why this is happening. Without the surprise signal, the goal proposal network does not get enough positive training data to learn from; hence, constantly samples random goals prohibiting successful switches which would create additional training data. Logically, the resource allocation speeds up learning such that the hard tasks are mastered faster, cf. CWYC† and CWYC.

## 6  Conclusion

We present the *control what you can* (CWYC) method that makes an autonomous agent learn to control the components of its environment effectively. We adopt a task-planning agent architecture while all components are learned from scratch. Driven by learning progress, the IM agent learns an automatic curriculum which allows it to not invest resources in uncontrollable objects, nor try unproportionally often to improve its performance on not fully solvable tasks. This key feature differentiates CWYC from approaches solely based on curiosity.

## Acknowledgement

Jia-Jie Zhu is supported by funding from the European Union's Horizon 2020 research and innovation programme under the Marie Skłodowska-Curie grant agreement No 798321. The authors thank the International Max Planck Research School for Intelligent Systems (IMPRS-IS) for supporting Sebastian Blaes and Marin Vlastelica Pogančić. We acknowledge the support from the German Federal Ministry of Education and Research (BMBF) through the Tübingen AI Center (FKZ: 01IS18039B).

## Footnotes

[1]https://github.com/n0c1urne/hrl.git

[2]an unsuccessful rollout takes 1600 steps, so $10^7$ steps are in the order of $10^4$ trials

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
