[Supplementary Material · appendix.pdf]

# Supplementary Material to
# Control What You Can: Intrinsically Motivated Task-Planning Agent

The supplementary information is structured as follows. We start with the algorithmic details in the next section and elaborate on the goal proposal networks. We provide the pseudocode in Sec. B and further information on the environment in Sec. C. In Sec. D we discuss the oracle baselines followed by Sec. E providing further analysis of the ablation studies. Finally we report the parameters for the architectures and the training in Sec. F.

## A    Details of the method

### A.1    Final task selector

The task selector $\mathcal{T}$ [3] models the learning progress when attempting to solve a task and is implemented as a multi-armed bandit. The reward is given in Eq. 1. We use the absolute value of the learning progress $|\rho|$ because the system should both learn when it can improve, but also if performance degrades [2]. Initially, the surprise term dominates the quantity. As soon as actual progress can be made $\rho$ takes the leading role. The reward is non-stationary and the action-value is updated according to

$$Q^{\mathcal{T}}(i) = Q^{\mathcal{T}}(i) + \alpha^{\mathcal{T}}(r^{\mathcal{T}}(i) - Q^{\mathcal{T}}(i)) \tag{S1}$$

with learning rate $\alpha^{\mathcal{T}}$. The task selector is to choose the (final) task for each rollout relative to their value accordingly. We want to maintain exploration, such that we opt for a stochastic policy with $p^{\mathcal{T}}(\tau = i) = Q^{\mathcal{T}}(i)/\sum_j Q^{\mathcal{T}}(j)$.

### A.2    Low-level control

Each task $i$ has its own policy $\pi_i$ which is trained separately using an off-policy deep RL algorithm. We use soft actor critic (SAC) [4] in the synthetic environment and DDPG+Her [1] in the robotics environment. Policies and the critic networks are parametrized by the goal (UVFA [6]).

### A.3    Subgoal sampling

For each subtask the goal is selected with the maximal value in the attention map. However, coordinates of tasks that are still to be solved in the task-chain are fixed, because they can likely not be controlled by the current policy. Formally:

$$s^* = \arg\max_{s'} \quad G_{i,j}(s') \tag{S2}$$

$$\text{subject to } s'_{m_\tau} = s_{m_\tau}, \ \forall \tau \in \kappa(i+)$$

where $\kappa$ is the task-chain and $\kappa(i+)$ denotes all tasks after $i$ and including $i$. $s_{m_\tau}$ selects the coordinates belonging to task $\tau$, see Sec. 3.1. The goal for subtask $j$ is then $g_j(s) = s^*_{m_j}$. This is a convex program and its solution can be computed analytically.

### A.4 Intrinsic motivations

For computing the success rate we use a running mean of the last $Z = 10$ attempts of the particular task:

$$\mathrm{sr}_i = \mathbf{1}/z \sum_{z=0}^{Z} \mathrm{succ}_i(-z) \tag{S3}$$

where $\mathrm{succ}_i(-z) \in [0, 1]$ denotes the success in the $z$-th last rollout where task $i$ was attempted to be solved.

The learning progress $\rho_i$ is then given as the finite difference of $\mathrm{sr}_i$ between subsequent attempts of task $i$.

To compute the surprise signal $\mathrm{surprise}_i$, we compute the statistics of the prediction error over all the collected experience, i.e. we assume

$$(e_i(t) - e_i(t-1)) \sim \mathcal{N}(\mu_i, \sigma_i^2) \tag{S4}$$

and compute the empirical $\mu$ and $\sigma$. Denoting the finite difference by $\dot{e}_i$, surprise within one rollout is then defined as

$$\mathrm{surprise}_i(t) = \begin{cases} 1 & \text{if } |\dot{e}_i(t)| > \mu_i + \theta \cdot \sigma_i \\ 0 & \text{otherwise.} \end{cases} \tag{S5}$$

where $\theta$ is a hyperparameter that needs to be choose.

### A.5 Training details of the goal proposal network

In an ever-changing environment as the ones presented in this paper, the goal proposal networks are a critical component of our framework that aim to learn relations between entities in the world. Transitions observed in the environment are labeled by the agent in interesting and undetermined transitions. Interesting transitions are those, in which a surprising event (high prediction error) occurs or which lead to an success in task $i$ given some other task $j$ was solved before, see Eq. 4. All other transitions are labeled as undetermined, since they might contain transition which are similar to those that are labeled interesting but didn't spark high interest. Coming back to our running example: bumping into, hence suddenly moving, the tool might spark interest in the tool because of a suddenly jump in prediction error. In general, the behaviour of an object after the surprising event is unknown and label for these transitions is not clear. Conclusively, we discard all undetermined transition within a rollout that come after a transition with positive label.

After removing all data that might prevent the goal proposal networks from learning the right relations it remains the problem that positive events are rare compared to the massive body of undetermined data. Hence, we balance the training data in each batch during training.

To make efficient use of the few positive samples we collect in the beginning of the training we impose a structural prior on the goal proposal network given by Eq. 3. The weight matrices are depicted in Fig. S1. This particular structure restricts the hypothesis space of the component to positional relations between components in the observation space that contains entities in the environment. In the main text, Figure 7 shows a compact representation of the initial and final weight matrices for different tasks that are computed by taking the minimum over $|w^1|$ (left column) and $|w^2|$ (middle column) in Fig. S1.

To understand the parametrization, consider to model that two components $(k, l)$ of $s$ should have the same value for a possitive signal, then $w_{kl}^1 \approx -w_{kl}^2$ should be nonzero and $w_k^3 = 0$. In this case the corresponding term in the exponent of Eq. 4 is zero if $s_l = s_k$. We see that in the case of the learned $G$ in Fig. S1 this relationship is true for the relevant components (position of agent, tool and object).

### A.6 Training / overall procedure

All components of CWYC start in a complete uninformed state. A rollout starts by randomly scramble the environment. The (final) task is chosen by the task selector. The task planner constructs the task chain $\kappa$. Every 5 steps in the environment, the goal proposal networks computes a goal for the

Figure S1: Weights learned by goal proposal networks for different task transitions. The left column shows the weights of $w^1$, the middle column of $w^2$ and the right column of $w^3$ (see Eq. 3).

current task. Given the subgoal the goal-parametric policy of that task is used. Whenever the goal is reached (up to a certain precision) a switch to the next task occurs. Again the goal proposal network is employed to select a goal in this task, unless it is the final task where the final goal is obviously used. If a goal cannot be reached the task ends after $T_T$ steps. In practice we run 5 rollouts in parallel. Then all components are trained using the collected data. For the task selector and task planner we use Eq. S1 and Eq. 2, respectively. Forward model and $G$s are trained using square-loss and Adam [5]. The policies are trained according to SAC/DDPG+HER. Pseudo-code and implementation details can be found in Sections (B, F).

# B  Pseudocode

The pseudocode for the method is given in Algorithm 1.

# C  Environments

## C.1  Synthetic environment

The synthetic environment is depicted in Fig. 2a and is simulated by the physics engine MuJoCo. The agent is modeled by a ball that is controlled by applying force in the $x$ and $y$ axis, so the agent's action corresponds to a 2-dimensional vector:

$$a = (F_x, F_y) \tag{S6}$$

**Algorithm 1** CWYC

```
 1: for episode in episodes do
 2:     sample main task τ_final ∼ T
 3:     sample main goal g_{τ_final} from environment
 4:     compute task chain κ using B starting from τ_final
 5:                                                                    // κ contains list task indices
 6:     i = 1
 7:     while t < T^max and no success in τ_final do
 8:         τ = τ_{κ[i]}
 9:         if τ ≠ τ_final then
10:             sample goal g_τ from G_{κ[i],κ[i+1]}                    // Eq. S2
11:         end if
12:         try to reach g_τ with policy_τ
13:         if succ_{κ[i]} then
14:             i = i + 1                                              // next task in task chain
15:         end if
16:     end while
17:     store episode in history buffer
18:     calculate statistics based on history
19:     train policies for each task
20:     train B                                                       // Sec. 3.2
21:     train all G                                                   // Sec. 3.2
22:     train T                                                       // Sec. 3.2
23: end for
```

The motion of the agent is subject to the laws of motion with the application of friction from the environment which makes it non-trivial to control. Other than the agent, the environment contains objects with different dynamics. The positions of the objects are part of the observation space of the agent along with a flag that specifies if the object has been picked up by the agent. We are dealing with a fully observable environment.

We define the goal spaces of the tasks as corresponding to the position of the individual objects. Some objects are harder to move than others and have other objects as dependencies. This means that the agent has to find this relation between them in order to successfully master the environment.

The types of objects that are used in the experiments are the following:

- *Static objects* cannot be moved

- *Random objects* move randomly in the environment, but cannot be moved by the agent

- *50% light objects* can be moved in 50% of the rollouts

- *Tool* can be moved and used to move the heavy object

- *Heavy objects* can be moved when using the tool

The observation vector for $d$ objects is structured as follows $(x, y, o_x^1, o_y^1, \ldots, o_x^d, o_y^d, \dot{x}, \dot{y}, p^1, \ldots, p^d)$, where $(x, y)$ is the position of the agent, $(o_x^i, o_y^i)$ is the position of the $i$-th object and $p^i$ indicates whether the agent is in possession of the $i$-th object. The goal spaces are the coordinates of the agent $(x, y)$ and the coordinates of each object $(o_x^i, o_y^i)$.

## C.2 Robotic environment

The robotic environment is depicted in Fig. 2c. The state space is 40 dimensional. It consists of the agent position and velocity, the gripper state, the absolute and relative positions of the box and the hook, respectively, as well as their velocities and rotations.

The environment is based on the OpenAI gym [3] PickAndPlace-v1 environment.

Final goals for the reach and tool task are sampled close to the initial gripper and tool location, respectively. Final goals for the object task are spawned in close proximity to the initial box position

Figure S2: Oracle dependency graph.

such that the box needs to be pulled closer to the robot but never pushed away. The box is spawned in close proximity to (closer to the robot) the upper end of the hook.

## D Oracle baselines

### D.1 CWYC with oracle goals

To assess the maximum performance of CWYC in the described settings, we crafted an upper baseline in which all learned high-level components, except for the final task selector $\mathcal{T}$, are fixed and set to their optimal value.

In the distractor setting, every task is solved by first doing the locomotion task. The goal proposal network $G_{i,j}(s)$ returns always the state value $s_{m_i}$, reflecting the ground truth relation we try to learn.

In the synthetic tool-use setting, the task graph depicted in Figure S2 is used. The goal proposal network $G_{i,j}(s)$ returns always the state value $s_{m_i}$, reflecting the ground truth relation we try to learn.

### D.2 HIRO/SAC with oracle reward

To see if HIRO manages to solve the synthetic environment at all, we constructed a oracle version of HIRO. The oracle receives as input not only the distance from, e.g., tool to target position but additionally the distance from agent to tool. This signal is rich enough to allow HIRO to solve the tool manipulation task as shown in Fig. 8(d) in the main text, although it still takes a lot of time compared to CWYC. We trained the SAC baseline on the same hybrid reward as well.

## E Additional analysis of the ablation studies

Without the surprise signal CWYC$^\star$ neither learns a meaningful resource allocation schedule, see Fig. S3(a), nor a task dependency graph, see Fig. S3(b). This highlights again the critical role of the surprise signal.

## F Training Details and Parameters

### F.1 Synthetic environment

- **Training:**
  # parallel rollout workers:    5

|          | start | loco | tool | heavy obj | 50% obj | random |
|----------|-------|------|------|-----------|---------|--------|
| loco     | 1.0   | 0.0  | 0.0  | 0.0       | 0.0     | 0.0    |
| tool     | 0.2   | 0.3  | 0.0  | 0.2       | 0.2     | 0.2    |
| heavy obj| 0.2   | 0.2  | 0.2  | 0.0       | 0.2     | 0.2    |
| 50% obj  | 0.2   | 0.2  | 0.2  | 0.2       | 0.0     | 0.2    |
| random   | 0.2   | 0.2  | 0.2  | 0.2       | 0.2     | 0.0    |

Figure S3: (a) Ressource allocation and (b) task dependency graph for the ablated version CWYC⋆. In (a) all tasks except locomotion behave identically because no progress is made.

- **Environment:**

| arena size: | $20 \times 20$ |
|---|---|
| $T^{\max}$: | 1600 |
| $\delta$: | 1.0 |

- **SAC:**

| lr: | $3 \times 10^{-4}$ |
|---|---|
| batch size: | 64 |
| policy type: | gaussian |
| discount: | 0.99 |
| reward scale: | 5 |
| target update interval: | 1 |
| tau (soft update) | $5 \times 10^{-3}$ |
| action prior: | uniform |
| reg: | $1 \times 10^{-3}$ |
| layer size $(\pi, q, v)$: | 256 |
| # layers $(\pi, q, v)$: | 2 |
| # train iterations: | 200 |
| buffer size: | $1 \times 10^{6}$ |

- **Forward model:**

| lr: | $10^{-4}$ |
|---|---|
| batch size: | 64 |
| input: | $(o_{t-1}, u_{t-1})$ |
| confidence interval: | 5 |
| network type: | MLP |
| layer size: | 100 |
| $\theta$: | 5 |
| # layers: | 9 |
| # train iterations: | 100 |

- **Final task selector:**

| $\beta^{\mathcal{T}}$: | $10^{-1}$ |
|---|---|
| lr: | $10^{-1}$ |
| random_eps: | 0.05 |
| surprise history weighting: | 0.99 |

- **Task planner:**

| $\beta^{B}$: | $10^{-3}$ |
|---|---|
| avg. window size: | 100 |
| surprise history weighting: | 0.99 |
| sampling_eps: | 0.05 |

- **Goal proposal network:**

|  |  |
|---|---|
| lr: | $10^{-4}$ |
| batch size: | 64 |
| L1 reg.: | 0.0 |
| L2 reg.: | 0.0 |
| $\gamma$ init: | 1.0 |
| $\gamma$ trainable: | True |
| # train iterations: | 100 |

## F.2 Robotic environment

- **Training:**

  | | |
  |---|---|
  | # parallel rollout workers: | 5 |

- **Environment:**

  | | |
  |---|---|
  | $T^{\text{max}}$: | 150 |
  | $\delta$: | 0.05 |

- **DDPG+HER:**

  | | |
  |---|---|
  | Q_lr: | $10^{-3}$ |
  | pi_lr: | $10^{-3}$ |
  | batch size: | 256 |
  | polyak | 0.95 |
  | layer size $(\pi, q)$: | 256 |
  | # layers $(\pi, q)$: | 3 |
  | # train iterations: | 80 |
  | buffer size: | $1 \times 10^{6}$ |
  | action_l2: | 1.0 |
  | relative goals: | false |
  | replay strategy: | future |
  | replay_k: | 4 |
  | random_eps: | 0.3 |
  | noise_eps: | 0.2 |

- **Forward model:**

  | | |
  |---|---|
  | lr: | $10^{-4}$ |
  | batch size: | 64 |
  | input: | $(o_{t-1}, u_{t-1})$ |
  | confidence interval: | 3 |
  | network type: | MLP |
  | layer size: | 100 |
  | $\theta$: | 3 |
  | # layers: | 9 |
  | # train iterations: | 100 |

- **Final task selector:**

  | | |
  |---|---|
  | $\beta^{\mathcal{T}}$: | $10^{-1}$ |
  | lr: | $10^{-1}$ |
  | random_eps: | 0.05 |
  | surprise history weighting: | 0.99 |

- **Task planner:**

  | | |
  |---|---|
  | $\beta^{B}$: | $10^{-3}$ |
  | avg. window size: | 100 |
  | surprise history weighting: | 0.99 |
  | sampling_eps: | 0.05 |

- **Goal proposal network:**

| | |
|---|---|
| lr: | $10^{-4}$ |
| batch size: | 64 |
| L1 reg.: | 0.0 |
| L2 reg.: | 0.0 |
| $\gamma$ init: | 1.0 |
| $\gamma$ trainable: | True |
| # train iterations: | 30 |