[Reviews · NeurIPS 2019]

Reviewer 1



Originality: The problem studied in this paper ("how to make an autonomous agent learn to gain maximal control of its environment") is not novel, with much prior work (see [Klyubin 2005] for a good overview). While the algorithm itself is novel, it seems like a bit of an ad hoc combination of ideas: bandits, curiosity, goal-conditioned RL, planning. Said another way, all sufficiently complex methods are likely to be original, but that in no way guarantees that they will be useful. Quality: While I laud the authors for including some baselines in their experiments, I think they were remiss to not include any intrinsic motivation baselines or planning baselines. One simple baseline would be to use SAC to maximize reward + surprise. Exploration bonuses such as RND [Burda 2018] and ICM [Pathak 2017] are good candidates as well. Given that the environments tested are relatively simple, I expect planning algorithms such as iLQR or PRMs could work well. Clarity: In terms of experiments, the method is complex enough that I don't think I'd be able to reproduce the experiments without seeing code. Code was not provided. The writing is vague in a number of places, and contains many typos (see list below). Significance: I agree with the authors that we must develop control algorithms that work well when provided only limited supervision. However, my impression is that this method actually requires quite a bit of supervision, first in specifying the goal spaces, again in defining the distance metric for each (L121), again in defining the procedure for sampling from each, and finally for tuning hyperparameters. I suspect that a method with 8 components has a large number of hyperparameters.

Reviewer 2



This paper introduces an algorithm to learn how to execute tasks that might depend on other tasks being executed first. Each task is a goal-conditioned problem where the agent needs to bring a specific part (coordinates) of the observed state to a given target. The challenging component is that, to modify some coordinates of the state (like the position of a heavy object), the agent first needs to modify other coordinates (move itself to a tool, then bring the tool to the heavy object). Therefore, methods that always set goals in the full state-space like HER or HIRO don’t leverage anyhow the underlying structure and might considerably struggle with this environments. This is an important problem to solve, and the assumptions are reasonable. Instead, the algorithm the authors propose fully leverages this structure, even if they only assume that the coordinates split is given (not their ordering or relationship). - First they have a “task selector” that picks which of the coordinate sets (ie, task) to generate a random goal in. This task selector is trained as a bandit with a learning progress reward. - The goal is then passed to a subtask planner that decides if another task should be executed before. To do so, a directed dependency graph of all task is built, where the weight on the edge between task i and j is the success on executing task i when j was exectued before (in previous attempts). The outgoing edges from node i are normalized, and an epsilon-greedy strategy is used to sample previous tasks. - Once the sub-task to execute is decided, a “goal proposal network” gives what were the previously successfull transition states. This network is regressed on previous rollouts, with some batch balancing to upweight the positive examples. - Finally, the actual goal is passed to a policy that is trained with SAC or HER. All points above also use a surprise component that seems to be critical for this setup to perform well. For that, a forward model is learned, and when the prediction error exceeds a fixed threshold the event is considered surprising. I would say that each individual piece is not drastically novel, but this is the first time I see all the components working successfully together. The clarity of the method explanation could be improved though. Their results are quite impressive, even compared to other hierarchical algorithms like HIRO (albeit none of their baselines leverages all the assumptions on the task structure as they do). They claim that learning progress is better than prediction error as intrinsic motivation (IM) because “it renders unsolvable tasks uninteresting as soon as progress stalls”. This is not convincing, because even in the cases they describe where an object can’t be moved, a model-error driven IM will stop trying to move it once it understands it can’t be moved. Furthermore, it seems from their results that only using the surprise, and no other type of IM is already almost as good. I would argue that the strength of their method is in the architecture, rather than in one IM or another. It might be better to tune down a bit the claims on which IM is better. Overall, it is a slightly involved architecture, but that powerfully leverages the given assumptions. The paper needs some serious proof-reading, and the presentation and explanations could also be drastically improved. Still, I think this is a good paper, with good enough experiments, analysis and ablations. It would be interesting to push this algorithm to its limit, by increasing the number of tasks available, making the task graph more challenging than a simple chain, adding more stochasticity to the environment, etc.

Reviewer 3



Note: the website links are switched around ("code" links to video). This paper proposes a method for intrinsic motivation in hierarchical task environments by combining surprise-based rewards with relational reasoning. The problem that is addressed is an important one, as RL often fails in long horizon, multistep tasks. The idea of "controlling what you can" is an intuitive and appealing approach to thinking about these kinds of environments. CWYC assumes that the state space has been partitioned into different tasks. This assumption is not entirely unreasonable given the power of object detectors and localization. Method: CWYC has many components that optimize different objectives. The task selector component is maximizing learning progress. A task graph learns which tasks are dependent on which, and selects a path through the graph. Subgoals are generated for each subtask with an attention process over all states, with different parameters per task. Goal conditioned policies for each task. Originality: No single component in this method is orignal, but their combination is. I found it to be an interesting way to use both intrinsic motivation and object relations. This method seems to provide a very good curriculum for learning complex tasks. Quality: The experiments are well documented and the method outperforms other methods, but it does not compare to other intrinsic motivation methods. Clarity: Good. Significance: Overall, I find the paper significant. While the assumption of a partitioned state space is a lot, it is not unreasonable with tools such as object detectors. The goal proposal structured output should be discussed more in the main text. Potential Issues: - The task graph is not a function of the particular goal in the final task: this means that it may be unable to choose a good sequence of subtasks if the sequence should depend on the particular goal (if you want the heavy object to go left, you need tool 1, but to move it right, you need tool 2). - The subgoal attention requires attending over all possible goals, which could be a problem in continuous or high dimensional state spaces. - The experiments should compare to "Curiosity-driven Exploration by Self-supervised Prediction"

[Author Response · NeurIPS 2019]

We thank the reviewers for the thorough feedbacks. Based on those, we have made numerous improvements.

**Implement a new IM baseline: ICM (Pathak 2017 [23].**
**Original code is for decrete actions.)** As suggested by *re-*
*viewer #1, #3*, we have implemented ICM for the synthetic
environment (Sec.4, Fig. 3 of the manuscript). The ICM base-
line uses SAC with an augmented reward: $r_t = r_t^{\text{ex}} + \alpha r^{\text{in}}$,
where $r_t^{\text{ex}}$ is the extrinsic reward (negative distance to goal) and
$r_t^{\text{in}}$ is an intrinsic reward.

The first experiment (Fig. 1 Left) follows the original ICM,
where the intrinsic reward signal is given by the total predic-
tion error: $r^{\text{in}} = \sum_i e_i(t)$, where the sum is over all goal
spaces/coordinates. Furthermore, we adapted ICM to make use
of the surprise signals that have shown to be important in the
manuscript. Thus, in a second experiment (Fig. 1 Right), the

Fig. 1: Synthetic environment in Sec. 4. Left: predic-
tion error; right: surprise. $\alpha$ is a hyperparameter we
scanned for.

intrinsic reward is given by the surprise signal: $r^{\text{in}} = \max_i \text{surprise}_i(t)$, where max is over goal spaces. Despite
scanning the hyperparameter $\alpha$, both IM baselines perform poorly and only solve the locomotion task, see Fig. 1.
Despite the seemingly simple environment, a random encounter of objects in continuous control is rare, given an agent
with heavy mass and a large arena.

To address *rev. #2*'s concern over "object can't be moved, a model-error driven IM will stop", we first clarify that the
issue, in fact, lies with the "random object" (in Sec. 4), not an unmovable object. We further tested the above-mentioned
IM baseline with the random object. The plot is similar to "tool" in Fig. 1 and we omit it due to space constraints.

**Clarify novelty and main contributions** We agree that each individual component is not original, as we have clearly
indicated they are from task-motion planning, IM, RL communities. We have already given references in the manuscript
(including Klyubin and Battaglia's work(s) mentioned by *rev. #1*). But combining them to successfully solve the
continuous control and robot trajectory optimization problem is novel (cf. *rev. #3*, originality).

*Rev. #1* suggested that the environments could be solved by classic planning methods. If one has an environment model
with an analytically (or accurate numerical) gradient, iLQR(G) may (without guarantee) solve the nonlinear program
(NLP). We have discussed this and other planning ideas (e.g. PRM) in the related work section. However, this paper is
based on model-free RL to solve the robot trajectory optimization through contact. We demonstrated IM/RL can solve
this as an alternative to NLP/sampling-based planning. This is beyond the scope of existing works such as Klyubin et al.

It is true that our method shares certain points with the concept of empowerment. We would like to emphasize that the
structure that we proposed leads to more efficient learning while maintaining the idea of maximizing controllability.

Fig. 2: Baselines with reward-shaping

**Concerning the complexity of our method** We acknowledge that the original Fig. 1
suggests an overwhelming complexity due to the detailed break-down (we will simplify
this). In fact, our *inductive bias (c.f. [Tenenbaum (2011) "How to grow..." ]) has only
3 modules (not 8)*: the task selector, planner, and subgoal generator. All other modules
are common among RL algorithms. In the ablation studies, we demonstrated that every
component is required to solve the task/maintain data efficiency. To further validate
this claim, we report additional results in Fig. 2, where the baselines are able to learn
the tool task with a hand-engineered reward: $r_t = r_t^{\text{ex}} - \text{dist}(\text{agent-pos}_t, \text{tool-pos}_t)$.
Therefore, our method in fact removes this additional layer of supervision.

42 **Further improvements.** *Code is uploaded* to the website as given in the paper. Con-
43 cerning our argument for playfulness, see [Smith (2005) "The dev. of embodied..."; Ryan (2000) "Intrinsic and
44 extrinsic..."]. Regarding prediction error vs. learning progress: prediction error fails in stochastic environments, see
45 [Oudeyer (2007) "Intrinsic motiv. systems..."; Burda (2018) "Large-scale..."].

46 **Q:** subgoal attention requires attending over all possible goals...? **A:** Our specific form of the goal generation network
47 allows for a closed-form solution to compute the argmax of the function. **Q:** The task graph is not a function of the
48 particular goal in the final task...? **A:** True. A limitation of our current architecture. **Q:** Goals within one task have
49 different difficulty. **A:** True. Interesting future direction. **Q:** When is the transition between sub-tasks happening...? **A:**
50 Your understanding is correct. If the goal can not be reached, the rollout is terminated after the maximum timesteps per
51 rollout is reached. We clarify this.

52 All text errors or vague language will be fixed. We have addressed other review comments but omit reporting them here
53 due to the space constraint. We gratefully acknowledge your help in improving the work.

[Meta-Review · NeurIPS 2019]

The reviewers were in disagreement about this paper. All reviewers agreed that the method was overly complex, making it less likely that the work would have lasting impact. However, they generally agreed that the paper contained useful ideas and contributions. We encourage the authors to address the reviewers comments. In addition to the reviewers comments, it is worthwhile to note that the idea of a partition goal space has been used in the past (e.g. in the hindsight experience replay paper, contrary to R2's comment). We encourage the authors to make this connection to prior work. Further, we strongly encourage the authors to include a comparison to HIRO with the same goal space partitioning, to provide an apples-to-apples comparison regarding the problem assumptions.